# Play to Grade: Grading Interactive Coding Games as Classifying Markov Decision Process

## Abstract

Contemporary coding education often present students with the task of developing programs that have user interaction and complex dynamic systems, such as mouse based games. While pedagogically compelling, grading such student programs requires *dynamic* user inputs, therefore they are difficult to grade by unit tests. In this paper we formalize the challenge of grading interactive programs as a task of classifying Markov Decision Processes (MDPs). Each student's program fully specifies an MDP where the agent needs to operate and decide, under reasonable generalization, if the dynamics and reward model of the input MDP conforms to a set of latent MDPs. We demonstrate that by experiencing a handful of latent MDPs millions of times, we can use the agent to sample trajectories from the input MDP and use a classifier to determine membership. Our method drastically reduces the amount of data needed to train an automatic grading system for interactive code assignments and present a challenge to state-of-the-art reinforcement learning generalization methods. Together with Code.org, we curated a dataset of 700k student submissions, one of the largest dataset of anonymized student submissions to a single assignment. This Code.org assignment had no previous solution for automatically providing correctness feedback to students and as such this contribution could lead to meaningful improvement in educational experience.

## 1 Introduction

The rise of online coding education platforms accelerates the trend to democratize high quality computer science education for millions of students each year. Corbett (2001) suggests that providing feedback to students can have an enormous impact on efficiently and effectively helping students learn. Unfortunately contemporary coding education has a clear limitation. Students are able to get automatic feedback only up until they start writing interactive programs. When a student authors a program that requires *user interaction*, e.g. where a user interacts with the student's program using a mouse, or by clicking on button it becomes exceedingly difficult to grade automatically. Even for well defined challenges, if the user has any creative discretion, or the problem involves any randomness, the task of automatically assessing the work is daunting. Yet creating more open-ended assignments for students can be particularly motivating and engaging, and also help allow students to practice key skills that will be needed in commercial projects.

Generating feedback on interactive programs from humans is more laborious than it might seem. Though the most common student solution to an assignment may be submitted many thousands of times, even for introductory computer science education, the probability distribution of homework submissions follows the very heavy tailed Zipf distribution – the statistical distribution of natural language. This makes grading exceptionally hard for contemporary AI (Wu et al., 2019) as well as massive crowd sourced human efforts (Code.org, 2014). While *code as text* has proved difficult to grade, actually running student code is a promising path forward (Yan et al., 2019).

We formulate the grading *via playing* task as equivalent to classifying whether an ungraded student program – a new Markov Decision Process (MDP) – belongs to a latent class of correct Markov Decision Processes (representing correct programming solutions to the assignment). Given a discrete set of environments $\mathcal{E} = \{e_n = (\mathcal{S}_n, \mathcal{A}, R_n, P_n) : n = 1, 2, 3, ...\}$, we can partition them into $\mathcal{E}^\star$

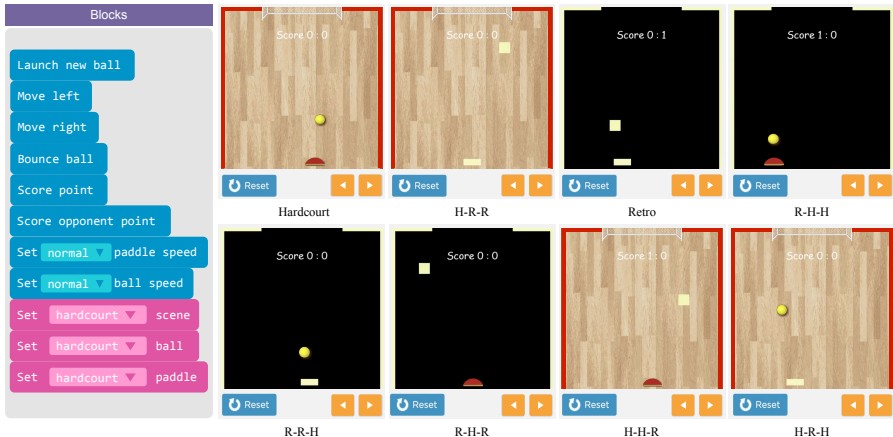

Figure 1: Bounce can have different "themes" for the background, paddle, and ball. There are two themes to choose from: "hardcourt" and "retro". We show the complete eight different combinations of themes and what the game would look like under these settings.

and $\mathcal{E}'$. $\mathcal{E}^\star$ is the set of latent MDPs. It includes a handful of reference programs that a teacher has implemented or graded. $\mathcal{E}'$ is the set of environments specified by student submitted programs. We are building a classifier that determines whether $e$, a new input decision process is behaviorally identical to the latent decision process.

Prior work on providing feedback for code has focused on text-based syntactic analysis and automatically constructing solution space (Rivers & Koedinger, 2013; Ihantola et al., 2015). Such feedback orients around providing hints and unable to determine an interactive program's correctness. Other intelligent tutoring systems focused on math or other skills that don't require creating interactive programs (Ruan et al., 2019; 2020). Note that in principle one could analyze the raw code and seek to understand if the code produces a dynamics and reward model that is isomorphic to the dynamics and reward generated by a correct program. However, there are many different ways to express the same correct program and classifying such text might require a large amount of data: as a first approach, we avoid this by instead deploying a policy and observing the resulting program behavior, thereby generating execution traces of the student's implicitly specified MDP that can be used for classification.

**Main contributions in this paper:**

- We introduce the reinforcement learning challenge of *Play to Grade*.

- We propose a baseline algorithm where an agent learns to play a game and use features such as total reward and anticipated reward to determine correctness.

- Our classifier obtains 93.1% accuracy on 8359 most frequent programs that cover 50% of the overall submissions and achieve 89.0% accuracy on programs that are submitted by less than 5 times. We gained 14-19% absolute improvement over grading programs via code text.

- We will release a dataset of over 700k student submissions to support further research.

## 2  THE PLAY TO GRADE CHALLENGE

We formulate the challenge with constraints that are often found in the real world. Given an interactive coding assignment, teacher often has a few reference implementations of the assignment. Teachers use them to show students what a correct solution should look like. We also assume that the teacher can prepare a few incorrect implementations that represent their "best guesses" of what a wrong program should look like.

To formalize this setting, we consider a set of programs, each fully specifies an environment and its dynamics: $\mathcal{E} = \{e_n = (\mathcal{S}_n, \mathcal{A}, R_n, P_n) : n = 1, 2, 3, ...\}$. A subset of these environments are

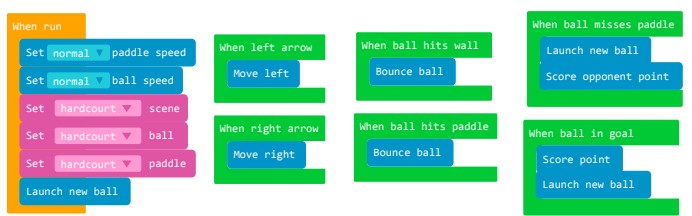 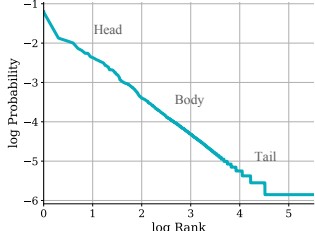

(a) We refer to this program as the **standard** program, as it is the simplest combination that makes up a correct program.

(b) The probability distribution conform to a Zipf distribution.

Figure 2: This is the drag-and-drop style code editor that students use to create the game Bounce. Conditions such as "when run" or "when left arrow" are fixed in space. Students place commands such as "score point", under each condition. The submission frequency of each unique program conforms to a Zipfian distribution on a log-log plot.

reference environments that are accessible during training, we refer to them as $\mathcal{E}^\star$, and we also have a set of environments that are specified by student submitted programs $\mathcal{E}'$. We can further specify a training set $\mathcal{D} = \{(\tau^i, y^i); y \in \{0, 1\}\}$ where $\tau^i \sim \pi(e^{(i)})$ and $e^{(i)} \sim \mathcal{E}^\star$, and a test set $\mathcal{D}_{\text{test}}$ where $e^{(i)} \sim \mathcal{E}'$. The overall objective of this challenge is:

$$\min \mathcal{L}(\theta) = \min_\theta \min_\pi \mathbb{E}_{e \sim \mathcal{E}}[\mathbb{E}_{\tau' \sim \pi(e)}[L(p_\theta(\phi(\tau', \pi)), y)]] \tag{1}$$

We want a policy that can generate trajectory $\tau$ that can help a classifier easily distinguish between an input environment that is correctly implemented and one that is not. We also allow a feature mapping function $\phi$ that takes the trajectory and estimations from the agent as intput and output features for the classifier. We can imagine a naive classifier that labels any environment that is playable (defined by being able to obtain rewards) by our agent as correct. A trivial failure case for this classifier would be that if the agent is badly trained and fails to play successfully in a new environment (returning zero reward), we would not know if zero reward indicates the wrongness of the program or the failure of our agent.

**Generalization challenge** In order to avoid the trivial failure case described above – the game states observed are a result of our agent's failure to play the game, not a result of correctness or wrongness of the program, it is crucial that the agent operates successfully under different correct environments. For any correct environment, $\mathcal{E}_+ = \{\mathcal{E}_+^\star, \mathcal{E}_+'\}$, the goal is for our agent to obtain the high expected reward.

$$\pi^\star = \arg\max_\pi \mathbb{E}_{e \sim \mathcal{E}_+}[\mathbb{E}_{\tau \sim \pi(e)}[R(\tau)]] \tag{2}$$

Additionally, we choose the state space to be the pixel-based screenshot of the game. This assumption imposes the least amount of modification on thousands of games that teaching platforms have created for students over the years.

This decision poses a great challenge to our agent. When students create a game, they might choose to express their creativity through a myriad of ways, including but not limited to using exciting background pictures, changing shape or color or moving speed of the game objects, etc. Some of these creative expressions only affect game aesthetics, but other will affect how the game is played (i.e., changing the speed of an object). The agent needs to succeed in these creative settings so that the classifier will not treat creativity as incorrect.

## 2.1 BOUNCE GAME SIMULATOR

We pick the coding game Bounce to be the main game for this challenge. Bounce is a block-based educational game created to help students understand conditionals[1]. We show actual game scenes in Figure 1, and the coding area in Figure 2a.

---

[1] https://studio.code.org/s/course3/stage/15/puzzle/10

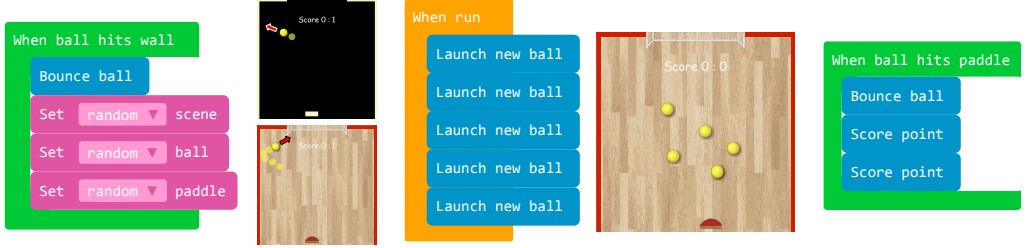

(a) Hit wall change to random theme     (b) Multiple balls when run     (c) Hit paddle wins two points

Figure 3: We demonstrate three examples of how Bounce can be programmed freely by allowing infinite repetition of commands and commands to be placed under any condition. Only (a) is considered correct since theme change does not affect game play. Both (b) and (c) are considered incorrect. (c) represents a reward design error (give points at incorrect condition). This demonstrates the difficulty of generalization in our setting.

The choice gives us three advantages. First, the popularity of this game on Code.org gives us an abundance of real student submissions over the years, allowing us to compare algorithms with real data. Second, a block-based program can be easily represented in a structured format, eliminating the need to write a domain-specific parser for student's program. Last, in order to measure real progress of this challenge, we need gold labels for each submission. Block-based programming environment allows us to specify a list of legal and illegal commands under each condition which will provide perfect gold labels.

The Bounce exercise does not have a bounded solution space, similar to other exercises developed at Code.org. This means that the student can produce arbitrarily long programs, such as repeating the same command multiple times (Figure 3(b)) or changing themes whenever a condition is triggered (Figure 3(a)). These complications can result in very different game dynamics.

We created a simulator faithfully executes command under each condition and will return positive reward when "Score point" block is activated, and negative reward when "Socre opponent point" block is activated. In deployment, such simulator needs not be created because coding platforms have already created simulators to run and render student programs.

## 2.2 CODE.ORG BOUNCE DATASET

*Code.org* is an online computer science education platform that teaches beginner programming. They designed a drag-and-drop interface to teach K-12 students basic programming concepts. Our dataset is compiled from 453,211 students. Each time a student runs their code, the submission is saved. In total, there are 711,274 submissions, where 323,516 unique programs were submitted.

**Evaluation metric** In an unbounded solution space, the distribution of student submissions incur a heavy tail, observed by Wu et al. (2019). We show that the distribution of submission in dataset conforms to a Zipf distribution. This suggests that we can partition this dataset into two sections, as seen in Figure 2b. **Head + Body**: the 8359 most frequently submitted programs that covers 50.5% of the total submissions (359,266). This set contains 4,084 correct programs (48.9%) and 4,275 incorrect programs (51.1%). **Tail**: This set represents any programs that are submitted less than 5 times. There are 315,157 unique programs in this set and 290,953 of them (92.3%) were only submitted once. We sample 250 correct and 250 incorrect programs uniformly from this set for evaluation.

**Reference programs** Before looking at the student submitted programs, we attempted to solve the assignment ourselves. Through our attempt, we form an understanding of where the student might make a mistake and what different variations of correct programs could look like. Our process can easily be replicated by teachers. We come up with **8 correct reference programs** and **10 incorrect reference programs**. This can be regarded as our training data.

**Gold annotations** We generate the ground-truth gold annotations by defining legal or illegal commands under each condition. For example, having more than one "launch new ball" under "when run" is incorrect. Placing "score opponent point" under "when run" is also incorrect. Abiding by this logic, we put down a list of legal and illegal commands for each condition. We note that, we intentionally chose the bounce program as it was amenable to generating gold annotations due to the API that code.org exposed to students. While our methods apply broadly, this gold annotation system will not scale to other assignments. The full annotation schema is in Appendix A.5.

## 3 RELATED WORK

**Education feedback** The quality of an online education platform depends on the feedback it can provide to its students. Low quality or missing feedback can greatly reduce motivation for students to continue engaging with the exercise (O'Rourke et al., 2014). Currently, platforms like Code.org that offers block-based programming use syntactic analysis to construct hints and feedbacks (Price & Barnes, 2017). The current state-of-the-art introduces a method for providing coding feedback that works for assignments up approximately 10 lines of code (Wu et al., 2019). The method does not easily generalize to more complicated programming languages. Our method sidesteps the complexity of static code analysis and instead focus on analyzing the MDP specified by the game environment.

**Generalization in RL** We are deciding whether an input MDP belongs to a class of MDPs up to some generalization. The generalization represents the creative expressions from the students. A benchmark has been developed to measure trained agent's ability to generalize to procedurally generated unseen settings of a game (Cobbe et al., 2019b;a). Unlike procedurally generated environment where the procedure specifies hyperparameters of the environment, our environment is completely determined by the student program. For example, random theme change can happen if the ball hits the wall. We test the state-of-the-art algorithms that focus on image augmentation techniques on our environment (Lee et al., 2019; Laskin et al., 2020).

## 4 METHOD

### 4.1 POLICY LEARNING

Given an observation $s_t$, we first use a convolutional neural network (CNN), the same as the one used in Mnih et al. (2015) as feature extractor over pixel observations. To accumulate multi-step information, such as velocity, we use a Long-short-term Memory Network (LSTM). We construct a one-layer fully connected policy network and value network that takes the last hidden state from LSTM as input.

We use Policy Proximal Optimization (PPO), a state-of-the-art on-policy RL algorithm to learn the parameters of our model (Schulman et al., 2017). PPO utilizes an actor-critic style training (Mnih et al., 2016) and learns a policy $\pi(a_t|s_t)$ as well as a value function $V^\pi(s_t) = \mathbb{E}_{\tau \sim \pi_\theta} \left[ \sum_{t=0}^{T} \gamma^t r_t | s_0 = s_t \right]$ for the policy.

For each episode of the agent training, we randomly sample one environment from a fixed set of correct environments in our reference programs: $e \sim \mathcal{E}_+^\star$. The empirical size of $\mathcal{E}_+$ (number of unique correct programs) in our dataset is 107,240. We focus on two types of strategies. One strategy assumes no domain knowledge, which is more realistic. The other strategy assumes adequate representation of possible combinations of visual appearances.

**Baseline training** : $|\mathcal{E}_+^\star| = 1$. We only train on the environment specified by the **standard** program displayed in Figure 2a. This serves as the baseline.

**Data augmentation training** : $|\mathcal{E}_+^\star| = 1$. This is the domain agnostic strategy where we only include the standard program (representing our lack of knowledge on what visual differences might be in student submitted games). We apply the state-of-the-art RL generalization training algorithm to augment our pixel based observation (Laskin et al., 2020). We adopt the top-performing augmentations (**cutout**, **cutout-color**, **color-jitter**, **gray-scale**). These augmentations aim to change colors

or apply partial occlusion to the visual observations randomly so that the agent is more robust to visual disturbance, which translates to better generalization.

**Mix-theme training**  : $|\mathcal{E}_+^\star| = 8$. This is the domain-aware strategy where we include 8 correct environments in our reference environment set, each represents a combination of either "hardcourt" or "retro" theme for paddle, ball, or background. The screenshots of all 8 combinations can be seen in Figure 1. This does not account for dynamics where random theme changes can occur during the game play. However, this does guarantees that the observation state $s$ will always have been seen by the network.

## 4.2 CLASSIFIER LEARNING

We design a classifier that can take inputs from the environment as well as the trained agent. The trajectory $\tau = (s_0, a_0, r_0, s_1, a_1, r_1, ...)$ includes both state observations and reward returned by the environment. We build a feature map $\phi(\tau, \pi)$ to produce input for the classifier. We want to select features that are most representative of the dynamics and reward model of the MDP that describes the environment. Pixel-based states $s_t$ has the most amount of information but also the most unstructured. Instead, we choose to focus on total reward and per-step reward trajectory.

**Total reward**  A simple feature that can distinguish between different MDPs is the sum of total reward $R(\tau) = \sum_t r_t$. Intuitively, incorrect environments could result in an agent not able to get any reward, or extremely high negative or positive reward.

**Anticipated reward**  A particular type of error in our setting is called a "reward design" error. An example is displayed in Figure 3(c), where a point was scored when the ball hits the paddle. This type of mistake is much harder to catch with total reward. By observing the relationship between $V^\pi(s_t)$ and $r_t$, we can build an N-th order Markov model to predict $r_t$, given the previous N-step history of $V^\pi(s_{t-n+1}), ..., V^\pi(s_t)$. If we train this model using the correct reference environments, then $\hat{r}$ can inform us what the correct reward trajectory is expected by the agent. We can then compute the hamming distance between our predicted reward trajectory $\hat{r}$ and observed reward trajectory $r$.

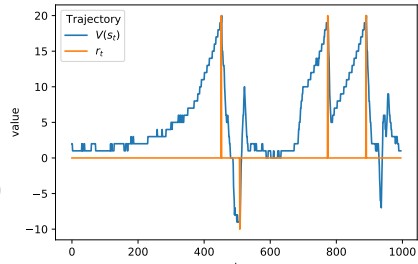

$$p(r_0, r_1, r_2...|\boldsymbol{v}) = p(r_0) \prod_{t=n}^{T} p(r_t|V^\pi(s_{<t}))$$

$$\approx p(r_0) \prod_{t=1}^{T} p(r_t|V^\pi(s_{t-n+1}), ..., V^\pi(s_t))$$

$$\hat{\boldsymbol{r}} = \arg\max_{\hat{\boldsymbol{r}}} p(\hat{\boldsymbol{r}}|\boldsymbol{v})$$

$$d(\boldsymbol{r}, \hat{\boldsymbol{r}}) = \text{Hamming}(\boldsymbol{r}, \hat{\boldsymbol{r}})/T$$

Figure 4: $V^\pi(s_t)$ indicates the model's anticipation of future reward.

**Code-as-text**  As a baseline, we also explore a classifier that completely ignores the trained agent. We turn the program text into a count-based 189-feature vector (7 conditions $\times$ 27 commands), where each feature represents the number of times the command is repeated under a condition.

## 5 EXPERIMENT

### 5.1 TRAINING

**Policy training**  We train the agent under different generalization strategy for 6M time steps in total. We use 256-dim hidden state LSTM and trained with 128 steps state history. We train each agent till they can reach maximal reward in their respective training environment.

**Classifier training**  We use a fully connected network with 1 hidden layer of size 100 and $\texttt{tanh}$ activation function as the classifier. We use Adam optimizer (Kingma & Ba, 2014) and train for

| K: # of trajectories sampled | K=1 | K=3 | K=5 | K=7 | K=8 |
|---|---|---|---|---|---|
| **Head + Body Student Programs** (360K submissions, 8359 unique) | | | | | |
| Code-as-text | 71.6 | — | — | — | — |
| Total Reward | 84.9 | 89.2 | 91.2 | 91.9 | 91.9 |
| Total Reward + Anticipated Reward | **85.7** | **90.7** | **92.7** | **93.5** | **94.1** |
| **Tail Student Programs** (350K submissions, 315K unique†) | | | | | |
| Code-as-text | 62.2 | — | — | — | — |
| Total Reward | 77.8 | 82.0 | 84.0 | 84.2 | 84.0 |
| Total Reward + Anticipated Reward | **81.2** | **86.4** | **88.0** | **88.8** | **89.0** |

Table 1: We evaluate the performance of our models on two sets of programs based on their frequency. †We sampled 500 programs uniformly from tail distribution for evaluation. With $K = 8$ trajectories per student our best model achieves **94.1%** accuracy on head body and **89.0%** on tail.

10,000 iterations till convergence. The classifier is trained with features provided by $\phi(\tau, \pi)$. We additionally set a heuristic threshold that if $d(\boldsymbol{r}, \hat{\boldsymbol{r}}) < 0.6$, we classify the program as having a reward design error.

To train the classifier, we sample 16 trajectories by running the agent on the 8 correct and 10 incorrect reference environments. We set the window size of the Markov trajectory prediction model to 5 and train a logistic regression model over pairs of $((V^\pi(s_{t-4}), V^\pi(s_{t-3}), ..., V^\pi(s_t)), r_t)$ sampled from the correct reference environments. During evaluation, we vary the number of trajectories we sample (K), and when $K > 1$, we average the probabilities over $K$ trajectories.

## 5.2 GRADING PERFORMANCE

We evaluate the performance of our classifier over three set of features and show in Table 1. Since we can sample more trajectories from each MDP, we vary the number of samples (K) to show the performance change of different classifiers. When we treat code as text, the representation is fixed, therefore $K = 1$ for that setting.

We set a maximal number of steps to be 1,000 for each trajectory. The frame rate of our game is 50, therefore this correspond to 20 seconds of game play. We terminate and reset the environment after 3 balls have been shot in. When the agent win or lose 3 balls, we give an additional +100 or -100 to mark the winning or losing of the game. We evaluate over the most frequent 8,359 unique programs that covers 50.1% of overall submissions. Since the tail contains many more unique programs, we choose to sample 500 programs uniformly and evaluate our classifier's performance on them.

We can actually see that using the trajectories sampled by the agent, even if we only have 18 labeled reference MDPs for training, we can reach very high accuracy even when we sample very few trajectories. Overall, MDPs on the tail distribution are much harder to classify compared to MDPs from head and body of the distribution. This is perhaps due to the fact the distribution shift that occurs for long-tail classification problems. Overall, when we add reward anticipation as feature into the classifier, we outperform using total reward only.

## 5.3 GENERALIZATION PERFORMANCE

One of our stated goal is for the trained agent $\pi$ to obtain high expected reward with all correct environments $\mathcal{E}_+$, even though $\pi$ will only be trained with reference environments $\mathcal{E}_+^\star$. We compare different training strategies that allow the agent to generalize to unseen dynamics. In our evaluation, we sample $e$ from the head, body, and tail end of the $\mathcal{E}_+$ distribution. Since we only have 51 environments labeled as correct in the head distribution, we evaluate agents on all of them. For the body and tail portion of the distribution, we sample 100 correct environments. The reward scheme is each ball in goal will get +20 reward, and each ball misses paddle gets -10 reward. The game terminates after 5 balls in or missing, making total reward range [-50, 100].

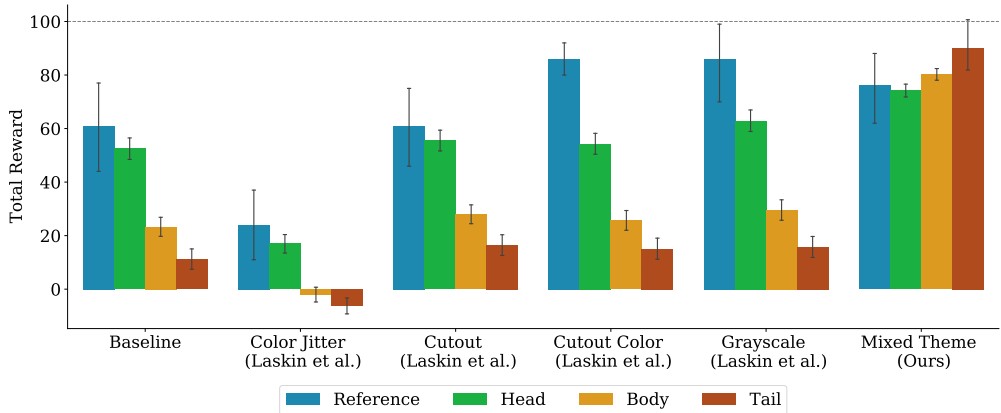

Figure 5: We compute the average reward across correct environments sampled from head, body, and tail of the distribution and we compare our **Mixed Theme** training to **Color Jitter**, **Cutout**, **Cutout Color**, **Grayscale** (Laskin et al., 2020). "Reference" represents a single environment specified by the **standard** program that all agents have been trained on. "Head" refers to the top 51 most frequently appeared correct programs. "Tail" refers to any correct programs that had less than 5 submissions in total. "Body" is every program in between. Total reward range is [-50, 100].

We show the result in Figure 5. Since every single agent has been trained on the reference environment specified by the **standard** program, they all perform relatively well. We can see some of the data augmentation (except Color Jitter) strategies actually help the agent achieve higher reward on the reference environment. However, when we sample correct environments from the body and tail distribution, every training strategy except "Mixed Theme" suffers significant performance drop.

## 6 DISCUSSION

**Visual Features**  One crucial part of the trajectory is visual features. We have experimented with state-of-the-art video classifier such as R3D (Tran et al., 2018) by sampling a couple of thousand videos from both the reference correct and incorrect environments. This approach did not give us a classifier whose accuracy is beyond random chance. We suspect that video classification might suffer from bad sample efficiency. Also, the difference that separates correct environments and incorrect environments concerns more about relationship reasoning for objects than identifying a few pixels from a single frame (i.e., "the ball going through the goal, disappeared, but never gets launched again" is an error, but this error is not the result of any single frame).

**Nested objective**  Our overall objective (Equation 1) is a nested objective, where both the policy and the classifier work collaboratively to minimize the overall loss of the classification. However, in this paper, we took the approach of heuristically defining the optimal criteria for a policy – optimize for expected reward in all correct environments. This is because our policy has orders of magnitude more parameters (LSTM+CNN) than our classifier (one layer FCN). Considering the huge parameter space to search through and the sparsity of signal, the optimization process could be very difficult. However, there are bugs that need to be intentionally triggered, such as the classic sticky action bug in game design, where when the ball hits the paddle through a particular angle, it would appear to have been "stuck" on the paddle and can't bounce off. This common bug, though not present in our setting, requires collaboration between policy and classifier in order to find out.

## 7 CONCLUSION

We introduce the Play to Grade challenge, where we formulate the problem of interactive coding assignment grading as classifying Markov Decision Processes (MDPs). We propose a simple solution to achieve 94.1% accuracy over 50% of student submissions. Our approach is not specific to the coding assignment of our choice and can scale feedback for real world use.

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
