# OpenReview forum: "Play to Grade: Grading Interactive Coding Games as Classifying Markov Decision Process"
_ICLR.cc/2021/Conference — Reject_

### Official Review · AnonReviewer3 · 2020-10-23
**This paper presents a class of interesting grading tasks, where the programs are interactive rather than functional. Despite the compelling problem statement, the paper ultimately under-delivers in what it actually does to solve this task.**

**Rating:** 4
**Confidence:** 4

**Review:**

- Quality : Okay
- Clarity : Poor
- Originality : Good at problem formulation, but then becomes bad at solution
- Significance : Could be very significant if done right (not quite there yet).

List of Cons :

A. Needs improvement on writing:

1)
The task of the students was confusing to understand. Specifically, for the first 2 pages I was completely lost on what the student is supposed to be implementing. I had the misconception that the coding task is something like Karel, where the student implement a program that is a policy, and are graded on how well the policy acted in a given game environment. That is not what this paper is about. I think the paper started with something like "Contemporary coding tasks in schools commonly ask the students to implement game environments, for instance, an assignment where the students implement the game PONG, and are graded on whether the resulting game plays like PONG. This types of assignment grading requires interaction as part of a grading process . . . " This would clear up the confusion.

2)
Equation 1) is unreadable. What does y = 0 mean? What is \tau ? What is \pi ? They are not explained before or after the equation with enough English sentences to intuitively explain what the loss function is doing. And as a result I cannot decipher this giant loss function that, supposedly when minimized according to \theta – what is \theta parameterizing? the game playing policy? the classifier of whether an environment is correct? – one would have solved the challenge of playing to grade.

3)
Equation 2) feels out of place, why all of a sudden we're maximizing reward on an environment? I thought we wanted to have the "most discerning policy". See C.1). The authors should make it clear that 2) is already a heuristic, and justifying why this concession is made up-front instead of putting it only in the discussion when the whole paper has ended: " However, in this paper, we took the approach of heuristically defining the optimal criteria for a policy – optimize for expected reward in all correct environments "

B. Needs better motivation of the challenges:

1)
I am not convinced that the grading policy should be run on pixel inputs. I think this is an unwarranted challenge. The authors consider changes to the perceptual input such as "random theme change can happen if the ball hits the wall", which seems highly limiting. What would happen if the student add a strobe effect that simply flashes the screen in all possible colors? What would happen if the student flips the world up-side-down? The authors remark that "Pixel-based states has the most amount of information but also the most unstructured." I agree, and I think one can easily resolve this issue by re-structuring the assignment such that the position of the balls and paddle can be easily extracted.

C. Needs better justification/explanation of proposed heuristic solution:

1)
Rather than training the agent to perform well on a game, one should be training agents that break the game in interesting ways, to discover bugs of the environment. For instance, the agent can first try to move the paddle past the left-side of the screen, one can imagine a faulty solution would allow the paddle to infinitely slide off the screen. This paper does not do this, instead, a heuristic of maximizing the reward on any given environment is proposed as a good proxy, due to the difficulty of jointly optimizing the classifier and the policy, again due to the fact of the authors choosing to use CNN as the policy representation.

List of Pros :
- Done great work at curating the dataset and make it runnable
- Done great work at specifying the problem statement, and developing a working (albeit misplaced) solution

Overall Comment:

This paper had great potential. It tackles a class of programs that otherwise would be very difficult to grade, because they require interactions. The problem statement of coordination between the policy and the loss function is beautiful, and it would've been great to see it being solved. However, the paper took a bad turn taking in pixel inputs for no apparent reason, despite owning everything in the simulation stack, from the source code to the rendering engine. This decision ultimately resulted in the paper not able to solve the coordinated optimization task; Resorted to a heuristic objective for the policy; Had to rely on a suite of data augmentation techniques to solve the problem of pixel input, an entirely artificial problem the authors created for themselves. A pixel level input is necessary when one is training RL agents that interacts with the real world where the "game engine" is unknown, but the students are literally implementing and handing the game engine to you to grade. Combined with poor writing quality, I think it is not yet ready for publication.

However this is a really good problem, and I would like to see this work being further developed, especially in solving the joint-optimization problem, where the agent can maximize for a "maximally informative trajectory" w.r.t. the classifier. Extending this approach to solve some additional domains of interactive programs would solidify this paper as one of the significant contributions to MOOCs grading line of literature.


final recommendations : I maintain my score. I think the other reviewer summarized it best, "interesting but immature". I hope to see this work develop and published in the future.

---

### Official Review · AnonReviewer1 · 2020-10-28
**Interesting work but immature**

**Rating:** 3
**Confidence:** 4

**Review:**

The authors contribute an approach to automatically distinguish between good and bad student assignment submissions by modeling the assignment submissions as MDPs. The authors hypothesize that satisfactory assignments modeled as MDPs will be more alike than they are to unsatisfactory assignments. Therefore this can potentially be used as part of some kind of future automated feedback system. The authors demonstrate this approach on an assignment for students to recreate a simple pong-like environment. They are able to achieve high accuracy over the most common submissions.

The major strength of the paper is in the novelty of the application domain. Reinforcement learning is not typically applied to the education domain in this way. I think there’s certainly potential. In addition, the results do give a positive signal about this research direction.

I have a number of concerns in terms of weaknesses of the paper. First, many of the details of the work are unclear in the current paper draft. I am not sure what the architecture for the agent and the classifier was exactly. Further I’m not sure what exactly the training data was for the agent and the classifier. Separately the authors include the numbers 711,274 (the number of submissions), 18 programs, and a single standard program. Finally it is unclear exactly how long the agents trained for, the authors only state that they trained the agents “until they can reach maximal reward”, without a sense of the number of training episodes.

I think this approach is very interesting, but I’m concerned about whether it’s an appropriate one for this problem domain. If the main issue is feedback then a RL approach is not going to be particularly helpful given RL’s blackbox nature. Further, since the authors could establish a gold standard baseline automatically it’s unclear why they would need an automated approach that only looked at pixels anyway (though it’s also unclear to me the extent to which the gold standard method used in this paper reflects how an instructor would evaluate this assignment). This is also a very simple environment which is well-suited to being modeled by an MDP (since it is a game), and I’m not convinced that this would scale to more complex environments. Finally, it’s unclear to me why student submissions would need to be evaluated from raw pixels and reward values. This seems like an arbitrary constraint since a CS instructor would have access to the model/MDP/submission directly.

I’m not convinced by the evaluation. The authors only compare to a relatively straightforward “code-as-text” baseline. While the authors mention that they attempted an image classifier and found that static images were not suitably discriminative, I don’t see why the authors couldn’t have tried sequences of images. I’d also like to see a more natural baseline, such as a hand-written classifier by a CS instructor or some approximation of this. Something as simple as a KNN classifier could also be worth considering for a baseline. Further, it’s unclear to me what the evaluation domains were, were they the 250 instances taken from the head and tail parts of the dataset? If so, were these all unique instances?

I think this work is interesting, but too immature for publication at this time. I’m recommending rejection.

Some questions for the authors:
-What was the training curriculum used for the agent(s)? Some clarity across the different numbers of training instances would be appreciated.
-Did the authors attempt to use sequences of frames in a more straightforward classifier? Or any other way to represent movement/video? Why not, if not?
-Is there something I’m missing about why this is an appropriate solution to the problem identified by the authors?

The language across the paper is also uneven and some claims are not substantiated. I’d recommend another pass. Some particular points:
- “teacher often has” -> “a teacher often has”
- Is it fair to assume teachers can prepare a few incorrect implementations? I’d like to see a citation to back up this claim, since my understanding is the opposite for complex assignments.
- “a simulator faithfully executes command”-> “a simulator that faithfully executes commands”
- “and will return positive reward when “Score point” ” -> “ and returns positive rewards when the “Score point” “
- “Socre opponent point”-> the “Score opponent point”
- “such simulator”-> “such a simulator”
- Code.org didn’t develop this interface they adapted it from Scratch (Maloney et al.)
- Can you present some evidence that your gold annotation is correct?
- Why is sample efficiency a concern?

---

### Official Review · AnonReviewer2 · 2020-10-28
**An interesting approach which could use more supporting experiments**

**Rating:** 5
**Confidence:** 4

**Review:**

The authors identify a relatively novel problem to solve with deep reinforcement learning - automatically grading programs which require dynamic input from the user to judge for implementation correctness. In doing so, they also introduce a new dataset and baseline benchmark which can be used to encourage further work in the area.

Some highlights:
- The work tries to take into account various real-world constraints of the problem. They have realistic expectations for what a teacher would have: one or more correct implementations and one or more incorrect ones. They deal with the possibility that students may choose to be creative and customize the visual aesthetic of the game in a multitude of ways and this must be addressed appropriately.
- They are methodical about defining the different parts of the evaluation distribution - Head + Body + Tail and the reasoning behind this.
- They use a few different methodologies for training based on changing the data distribution as well as the reward structure. This is a reasonable sort of ablation study.


Some limitations of the work:
- The work's only example game is 'Bounce' on Code.org. This is one of the most popular games and has an extremely large body of student submissions which give breadth to the training distribution. The programming language is 'block-based' which is extremely structured and distinctly more so than regular imperative programming languages.
- The annotation system used for Bounce will not scale to other problems.
- The authors acknowledge that there exists the challenge handling long-tailed distributions in their classification problem. This is probably going to be a significant hurdle for adapting this technique to other program languages/tasks where the tails are even longer. The task of grading is also something that should not have many false negatives (grading as incorrect when it was correct) since that's not good for students.

Some other comments:
- There appears to be a couple of grammar and one spelling-related typo in the paragraph immediately before the start of Section 2.2
- Right above section 2, consider changing the definition to  $\\mathcal{E}{+} = \\mathcal{E}^{\\star}{+} \cup \\mathcal{E}^{'}{+}$ (sorry couldn't figure out how to do the + subscript without markdown treating the underscores as signals for italicizing)
- In the paragraph right before Section 5.2, it would seem that there were 18 trajectories (rather than the stated 16) if you ran on 8 correct and 10 incorrect reference environments.

Overall, considering the three types of comments above, I believe that the paper is an interesting piece of work which makes some rather preliminary progress on a new methodology for automatically grading programming assignments which require dynamic input. However, it leaves some questions up in the air, namely how this approach would scale to different assignments. I would say 'Bounce' represents quite possibly the easiest programming assignment of this category, so it is not the most convincing. I would like to suggest doing the same sort of process for at least a couple of other games, even if they all use the same 'block programming' language from code.org. That would show how much by-hand engineering is required to address the long-tails, the different visual aspects of the game, the length and complexity of episodes, etc. Until then, I will have to say that this is below the threshold for acceptance.

---

### Decision · Program_Chairs · 2021-01-07
**Final Decision**

**Decision:**

Reject

**Comment:**

The paper studies a novel problem setting of automatically grading interactive programming exercises. Grading such interactive programs is challenging because they require dynamic user inputs. The paper's main strengths lie in formally introducing this problem, proposing an initial solution using reinforcement learning, and curating a large dataset from code.org. All reviewers generally appreciated the importance of the research problem studied and the potential of the work. Even though the reviewers found the work interesting, there was a clear consensus that the work is still immature and not yet ready for publication. I appreciate the authors' engagement with the reviewers during the discussion phase. Overall, the reviewers have provided very detailed and constructive feedback to the authors. We hope that the authors can incorporate this feedback when preparing future revisions of the paper.